# A Pilot Study of Heart Rate Variability Synchrony as a Marker of Intraoperative Surgical Teamwork and Its Correlation to the Length of Procedure

**DOI:** 10.3390/s22228998

**Published:** 2022-11-21

**Authors:** Katarzyna Powezka, Allan Pettipher, Apit Hemakom, Tricia Adjei, Pasha Normahani, Danilo P. Mandic, Usman Jaffer

**Affiliations:** 1Imperial Vascular Unit, Imperial College Healthcare NHS Trust, London W2 1NY, UK; 2Department of Electrical and Electronic Engineering, Imperial College London, London SW7 2AZ, UK; 3Department of Vascular Surgery, Imperial College Healthcare NHS Trust, London W2 1NY, UK

**Keywords:** heart rate variability, HRV synchrony, teamwork, length of operation

## Abstract

**Simple Summary:**

This is a single center prospective cross-sectional study, which showed that length of procedure is inversely correlated with heart rate variability synchronies of operating surgeons. Our work shows that HRV synchrony analysis is feasible and HRV synchronisation amongst operating surgeons can be used as an objective marker to quantify intraoperative teamwork.

**Abstract:**

Objective: Quality of intraoperative teamwork may have a direct impact on patient outcomes. Heart rate variability (HRV) synchrony may be useful for objective assessment of team cohesion and good teamwork. The primary aim of this study was to investigate the feasibility of using HRV synchrony in surgical teams. Secondary aims were to investigate the association of HRV synchrony with length of procedure (LOP), complications, number of intraoperative glitches and length of stay (LOS). We also investigated the correlation between HRV synchrony and team familiarity, pre- and intraoperative stress levels (STAI questionnaire), NOTECHS score and experience of team members. Methods: Ear, nose and throat (ENT) and vascular surgeons (consultant and registrar team members) were recruited into the study. Baseline demographics including level of team members’ experience were gathered before each procedure. For each procedure, continuous electrocardiogram (ECG) recording was performed and questionnaires regarding pre- and intraoperative stress levels and non-technical skills (NOTECHS) scores were collected for each team member. An independent observer documented the time of each intraoperative glitch. Statistical analysis was conducted using stepwise multiple linear regression. Results: Four HRV synchrony metrics which may be markers of efficient surgical collaboration were identified from the data: 1. number of HRV synchronies per hour of procedure, 2. number of HRV synchrony trends per hour of procedure, 3. length of HRV synchrony trends per hour of procedure, 4. area under the HRV synchrony trend curve per hour of procedure. LOP was inversely correlated with number of HRV synchrony trends per hour of procedure (*p* < 0.0001), area under HRV synchrony trend curve per hour of procedure (*p* = 0.001), length of HRV synchrony trends per hour of procedure (*p* = 0.002) and number of HRV synchronies per hour of procedure (*p* < 0.0001). LOP was positively correlated with: FS (*p* = 0.043; R = 0.358) and intraoperative STAI score of the whole team (*p* = 0.007; R = 0.493). Conclusions: HRV synchrony metrics within operating teams may be used as an objective marker to quantify surgical teamwork. We have shown that LOP is shorter when the intraoperative surgical teams’ HRV is more synchronised.

## 1. Introduction

A team can be defined as a “distinguishable set of two or more people who interact dynamically, interdependently, and adaptively toward a common and valued goal, objective or mission, who have each been assigned specific roles or functions to perform, and who have limited life span of membership” [1]. Teamwork in the operating theatre/room (OR) has been shown to be an essential contributor to patient safety [2]. It has been suggested that efficient teamwork depends on the ability of each team member to anticipate the needs of others and have shared understanding of how a given procedure should happen [3]. We have previously shown that team members’ familiarity may be a relevant factor shaping team collaboration and, thus, its effectiveness [4,5].

There are many factors that can adversely affect surgical cooperation, including failures in non-technical performance, which have been associated with higher rates of technical errors [6,7]. However, there is a need for an objective assessment tool that may be used to evaluate the effectiveness of intraoperative team working.

Excessive stress levels have a deleterious impact on technical performance of surgical teams, the effect depending on the expertise of the surgeon and the nature of the task [6]. The Imperial Stress Assessment Tool (ISAT), which incorporates physiological and subjective measures of intraoperative surgical stress, was developed to assess this for individual surgeons [8].

Heart rate variability (HRV) measures variations in intervals between consecutive heart beats (R–R intervals) and represents interplay between the sympathetic and parasympathetic nervous systems (SNS and PNS, respectively) in response to intrinsic and extrinsic factors. A significant correlation has been reported between HRV measurements recorded during complex surgical procedures and perceived intraoperative stress levels as evaluated by the STAI questionnaire [9].

Changes in HRV are commonly evaluated in two frequency bands: the low-frequency (LF) band, 0.04–0.15 Hz, which is linked to the interaction of the SNS and PNS, and the high-frequency (HF) band, 0.15–0.4 Hz, which primarily reflects the activity of the PNS. HRV synchrony can be defined as the degree to which operating surgeons’ HRV fluctuations track together [10]. Therefore, it may be useful for objective assessment of team cohesion. To the best of our knowledge, HRV synchrony among operating surgeons has not been previously investigated in the context of procedure outcome.

Our primary objective was to assess the feasibility of using intraoperative HRV synchrony to objectively quantify teamwork amongst surgeons involved in operations.

Our secondary objectives were:to explore which, if any, of the tested metrics for HRV synchrony (number of HRV synchronies per hour of a procedure (n-HRV-S), number of HRV synchrony trends per hour of a procedure (n-HRV-S_T_), length of HRV synchrony trends per hour of a procedure (L-HRV-S_T_) and area under the HRV synchrony trend peak per hour of a procedure (area-HRV-S_T_), correlated with length of procedure (LOP);to explore if the HRV synchrony metrics correlated with team familiarity (Familiarity Score, FS), preoperative and intraoperative stress levels (STAI score), experience of each surgeon and self-assessed non-technical skills (NOTECHS score);to assess whether intraoperative disruptions (“glitches”) are associated with any of the metrics of HRV synchrony.

## 2. Materials and Methods

Institutional approval to conduct this study was obtained prior to data collection. All participants were given written information regarding the study and consent was obtained. Vascular and ENT surgical team members were recruited into the study in the period from December 2016 to July 2017. During the procedure, continuous electrocardiogram (ECG) was recorded for every surgical team member. Surgical teams, which were included in the study, were composed of a consultant and two registrars. Each team member was asked about the level of their surgical experience. 

Self-reported stress was assessed with the validated short, six-item State Trait Anxiety Inventory for Adults (STAI) questionnaire, which was filled in before and after each procedure. The STAI questionnaire which was filled in before the procedure showed the preoperative stress level of each member of the surgical team and was labelled as preoperative STAI score. The STAI questionnaire which was filled in just after the procedure had finished showed levels of self-perceived intraoperative stress experienced by each surgical team member and was labelled as intraoperative STAI score. Total STAI score varies from 4 to 24, with higher scores indicating increased psychological stress [8,11].

Additionally, after the procedure, each surgeon self-assessed surgical team performance using the Oxford Non-Technical Skills (NOTECHS) questionnaire. Domains being assessed include leadership and management, teamwork and cooperation, problem-solving and decision-making and situation awareness. Each domain can be marked from 1 to 4, and the maximum score is 16 for each questionnaire [12,13,14]. The total mark for non-technical skills is the sum of the marks for each questionnaire and ranges from 12 to 48 for the surgical team composed of a consultant and two registrars.

Familiarity Score (FS) was calculated as the sum of the number of times that each possible pair of members of the team (vascular consultant, vascular registrars, ENT consultant, ENT registrars) within the team had worked together over the previous 6 months, divided by the number of possible combinations of pairs in the team [5].

Glitches (unexpected events which occurred during the procedure, that is, disruptions) were recorded according to already established groups by an independent observer with surgical experience, who was present during each procedure and documented the time of glitch occurrence [15]. We calculated glitch rate per hour of the procedure using the equation below:Glitch rate=(Glitch count ÷LOP)×60.

Experience of surgeons participating in the surgery (defined by years of work since surgical core training), length of procedure (LOP; time in minutes from the skin incision to the end of the skin closure) and Familiarity Score (FS) were recorded. Start time of recording as well as the start time of the procedure (knife to skin) were recorded separately for each surgeon.

Continuous ECG recording was performed using a custom-made portable amplifier [16]. Five ECG electrodes were placed on the chest under the scrubs of each surgeon at least fifteen minutes before the procedure in order to assess the HRV baseline of the individual: two electrodes on the left side of the sternum in the 3rd rib space—one on the midclavicular line and the second one on the anterior axillary line—and three on the right side in the 3rd rib space—one on the midclavicular line, one on the anterior axillary line and one on the midaxillary line. Each pair of surgeons had their heart rate variability analysed from continuous ECG recordings using the methodology described below.

### 2.1. Analysis of Heart Rate Variability (HRV) Synchronies

Heart rate variabilities of each surgeon were analysed using MATLAB software (R2015b_win64, Mathworks, Natick, MA, USA) at the Department of Electrical and Electronic Engineering, Imperial College, London.

The R-peaks from ECG recordings were extracted to derive HRV series. The R-R interval (time taken between two consecutive R-peaks) was used as a measure to calculate variability over time. As calculation of R-R intervals requires precision and accuracy, artefacts from wearable electrodes resulting from movements and imperfect electrode attachment were not analysed. The algorithm used in our study uses a combination of the Hilbert transform and matched filtering. Using the former ensures that the HRV series meets mono-component criteria (signal is a trend of a single value changing over time—such as the amplitude or frequency) [17]. 

A quantitative measure of the level of cooperation between surgeons’ physiological responses (HRV) was achieved based on the assessment of the phase relationship between multiple data channels using intrinsic phase synchrony (IPS) and intrinsic coherence under the framework referred to as intrinsic multiscale analysis [18]. The algorithms under this framework have the capability to quantify intra- and intercomponent dependences of a complex system such as multiple synchronies. For direct comparison, the signal can be decomposed using a process called empirical mode decomposition (EMD), which is essentially an adaptive, data-driven method for the analysis of non-linear and non-stationary time series [19]. It is used to decompose a given signal into its multiple narrow-band amplitude/frequency modulated components, which are referred to as intrinsic mode functions (IMFs) and are used as a basis for signal representation. Standard EMD algorithms, however, are highly sensitive to local signal vibrations, which in turn can result in the comparison of different bandwidths. Thus, for this analysis a variant was applied called noise-assisted multivariate empirical mode decomposition (NA-MEMD), which uses the artificial addition of noise to directly link the IMF channels together. 

IMFs representing the range of 0.04–0.4 Hz were selected, since this range of frequency is a combination of the low-frequency (LF) and high-frequency (HF) bands for parasympathetic and sympathetic nervous systems, respectively, and thus provide an indication for the involuntary reaction to events.

For collaborative tasks that take place over a longer period of time, it is more important to have a quantitative measure, which quantifies the level of cooperation in a bigger picture, i.e., trends in synchrony, rather than detailed frequencies. A proposed extension to IPS, which is referred to as nested intrinsic phase synchrony (N-IPS), employs the conventional phase relationship in IMFs, and further decomposes the resulting synchrony time series into multiple physically meaningful scales of synchrony where the scales of interest are then combined to generate the trend [10]. Performing direct comparison employed the calculation of the phase synchrony index (PSI), which ranges from zero to one where a greater value indicates greater synchrony. The difficulty of using the PSI, however, is in determining the statistical relevance of the calculated values. Thus, further noise which retains most of the properties of the comparative signals (except the phase locking) is applied to develop a baseline. Below this baseline, the synchrony is believed to be statistically insignificant, which implies no synchronisation. 

Trends in HRV synchrony between pairs of surgeons within the team were quantified using the phase synchrony index (PSI). Trends related to the PSI above baseline exhibit HRV synchronisation during procedures as well as physiological reactions to external events such as glitches during surgery [10] (Figure 1 and Figure 2). 

From the PSI, the following HRV metrics were derived: number of HRV synchronies (n-HRV-S) per hour of the procedure:
n−HRV−S=(Number of HRV synchronies÷LOP)×60,area under the HRV synchrony trend curves per hour (area-HRV-S_T_),length of HRV synchrony trends per hour (L-HRV-S_T_),number of HRV synchrony trends per hour (n-HRV-S_T_).

### 2.2. Statistical Analysis

Statistical analysis was conducted using SPSS (version 21, IBM, Chicago, IL, USA). Variables are presented as median and interquartile range (median, IQR). Correlations between variables were assessed using stepwise multiple linear regression. Significance was taken at the 95% level. Power calculations were not carried out for this feasibility study.

## 3. Results

### 3.1. Demographic Data

We analysed HRV synchronies in 24 surgical pairs, formed of 12 surgeons (vascular and ENT). Vascular procedures included open abdominal aortic aneurysm repair (*n* = 1 case), carotid endarterectomy (*n* = 3), femoro-crural artery bypass (*n* = 4), open lumbar sympathectomy (*n* = 1). ENT procedures included hemithyroidectomies (*n*=6), total thyroidectomies (*n* = 2), parathyroidectomy (*n* = 1) and endoscopic surgery (*n* = 1).

A detailed description of surgeons’ demographic data is presented in Table 1. 

Vascular procedures were longer than ENT procedures (median 208 min (IQR 114.25–251), median 84 min (IQR 73.5–92), respectively). Descriptive analysis of results is presented in Table 2. 

### 3.2. Feasibility Assessment

Five ECG electrodes and wearable devices were well tolerated by all participants and wearing them was not reported to interfere with their performance during the procedures. None of the approached surgeons refused participation in the study.

We recorded nineteen ENT procedures (57 pairs) and sixteen vascular procedures (48 pairs), which involved a consultant and two registrars in each speciality. Due to device disruption and ECG electrodes becoming detached during the procedure, leading to corruption of the recordings, we were only able to extract recorded heart rate in twelve ENT pairs and twelve vascular pairs.

### 3.3. Exploring Correlations between HRV Synchrony Metrics and Length of Procedure (LOP)

Stepwise multiple linear regressions were employed to analyse the relationship between HRV synchrony metrics and length of procedure. Each of the metrics of the HRV synchrony was analysed separately against LOP (Figure 3).

#### 3.3.1. Number of HRV Synchrony Trends per Hour (n-HRV-S_T_)

A multiple linear regression model predicted length of procedure (LOP) with statistical significance, F(6,17) = 10.098, *p* < 0.0001, adjusted R^2^ = 0.704. The independent variable, which was inversely associated with LOP with statistical significance, is n-HRV-S_T_: *p* < 0.0001 (Pearson’s correlation coefficient R = −0.752). The independent variables positively correlated with LOP with statistical significance: FS: *p* = 0.043 (R = 0.358), and the intraoperative STAI score of the whole team: *p* = 0.007 (R = 0.493). The whole team experience was not statistically significantly correlated with LOP: *p* = 0.068 (R = 0.313). 

#### 3.3.2. Area under the Peak of HRV Synchrony Trends per Hour (Area-HRV-S_T_)

A multiple linear regression model predicted LOP with statistical significance, F(6,17) = 6.813, *p* = 0.001, adjusted R^2^ = 0.603. The independent variable, which is inversely associated with LOP with statistical significance, was area-HRV-S_T_: *p* = 0.001 (Pearson’s correlation coefficient R = −0.595). Independent variables, that are positively correlated with LOP with statistical significance, included intraoperative STAI score of the whole team: *p* = 0.007 (R = 0.493) and FS: *p* = 0.043, (R = 0.358) (Figure 3). The whole team experience was not statistically significantly correlated with LOP: *p* = 0.068 (R = 0.313). 

#### 3.3.3. Length of HRV Synchrony Trends per Hour (L-HRV-S_T_)

A multiple linear regression model predicted LOP with statistical significance, F(6,17) = 4.682, *p* = 0.006, adjusted R^2^ = 0.490. The independent variable inversely correlated with LOP with statistical significance was L-HRV-S_T_: *p* = 0.002 (R = −0.574). The independent variables positively statistically significantly associated with length of procedure were the intraoperative STAI score for the whole team: *p* = 0.007 (R=0.493) and FS: *p* = 0.043 (R = 0.358). The whole team experience was not statistically significantly correlated with LOP: *p* = 0.068 (R = 0.313).

#### 3.3.4. Number of HRV Synchronies per Hour (n-HRV-S)

A multiple linear regression model predicted LOP with statistical significance, F(6,17) = 4.657, *p* < 0.0001, adjusted R^2^ = 0.448. n-HRV-S shows inversely statistically significant correlation with LOP: *p* < 0.0001 (R = −0.694). The intraoperative STAI score for the whole team and FS are positively associated with LOP with statistical significance: *p* = 0.007 (R = 0.493) and *p* = 0.043 (R = 0.358), respectively. The whole team experience was not statistically significantly correlated with LOP: *p* = 0.068 (R = 0.313). 

### 3.4. Exploring Correlations between HRV Synchrony Metrics and Surgical Teams’ Variables (FS, Preoperative and Intraoperative STAI Scores of Whole Team, NOTECHS Scores, Team Experience and Glitch Rate)

None of the created models in the stepwise linear regression manner was statistically significant, therefore correlations between HRV synchrony trends and FS, NOTECHS scores, glitch rate, preoperative and intraoperative STAI scores of the whole team and team experience could not be assessed.

### 3.5. Exploring Correlations between Glitches and LOP and HRV Synchrony Trends

Stepwise multiple linear regression was employed to explore correlations between glitch rate per hour and length of procedure. The created model showed that glitch rate per hour was not statistically significantly correlated with LOP: *p* = 0.902 (R = −0.144). Similar models showed that glitch rate per hour was not statistically significantly correlated with n-HRV-S: *p* = 0.207 (R = 0.188); L-HRV-S_T_: *p* = 0.533 (R = 0.054); area-HRV-S_T_: *p* = 0.835 (R = 0.125) and n-HRV-S_T_: *p* = 0.905 (R = 0.105).

### 3.6. Principal Component Analysis of HRV Synchrony Metrics

A principal component analysis (PCA) was run on four HRV synchrony metrics: L-HRV-S_T_, area-HRV-S_T_, n-HRV-S and n-HRV-S_T_. The suitability of PCA was assessed prior to analysis. Inspection of the correlation matrix showed that all variables had at least one correlation coefficient greater than 0.3. PCA revealed two components—L-HRV-S_T_ and n-HRV-S—that had eigenvalues greater than one and which explained 79.18% and 11.95% of the total variance, respectively.

## 4. Discussion

To the best of our knowledge, this is the first study which attempts to objectively assess team cooperation during surgical procedures using HRV synchrony. This builds on previous work our group has reported on HRV synchrony, detailing methodology for ECG data acquisition from wearable devices and intrinsic multiscale analysis [10,18]. Previous work suggested that HRV synchronisation can be used as an objective tool assessing team cohesion, which is considered to correlate with team performance [10].

We have shown that greater synchronisation of HRV within a surgical pair is associated with a shorter length of procedure. This is in keeping with previous work, which has reported that improved team function leads to better efficiency [20,21,22]. We identified four metrics of HRV synchrony, out of which L-HRV-S_T_ was shown to be the most predictive of LOP.

Given the previously reported work on team familiarity, we expected to find that greater team familiarity would be associated with shorter durations of surgery [5]. However, this was not demonstrated in this study. A potential explanation of this may involve overlap between HRV synchrony and team familiarity in the model. Further studies controlling for these factors potentially in a simulated environment may clarify.

Despite the available body of evidence indicating that non-technical skills influence surgical performance and, hence, indirectly impacting patient outcomes, we have not shown a significant correlation between the NOTECHS scores and the LOP [2,6,23]. The potential explanation could be a small sample size and further research should be performed to clarify that.

Interestingly, the collective team experience was not correlated with the LOP. It could be explained by our sample size that was small and, again, further research would clarify these findings for the whole teams as well as for the individual team members. 

We have found correlation between the intraoperative STAI scores of the whole team and length of the procedure. This strongly suggests that the higher perceived stress level can adversely impact the duration of the procedure. However, this is not the case with the preoperative STAI scores. The reason for that could be explained with further research. 

We did not show any relationship between glitch rate per hour of the procedure and LOP or HRV synchrony metrics. Again, this may be explained by small sample size, possible inaccuracy in observation when some glitches may have gone unnoticed and heterogeneity of surgical procedures, especially within vascular teams. We found that the majority of glitches were related to communication issues, distractions during the operation, theatre environment issues and preoperative planning.

With the development of objective HRV-based synchrony metrics, we hope to identify highly functioning teams as well as potentially weaker teams in a formative assessment setting. Being able to objectively measure performance enables us to develop strategies to mitigate the risk of poor team cooperation and train teams to work more effectively together.

## 5. Limitations

The main limitations of this study are the size of the sample and heterogeneity of the recorded procedures, especially within vascular teams. Additionally, our study focused only on the analysis of the surgical sub-team. Our ECG recording device was effective in recording, however, it was susceptible to failure due to movement and disconnection of electrodes. This can be mitigated in future studies by using “in-ear” recordings which our group has developed [24].

## 6. Conclusions

Acquisition of HRV recordings during surgical procedures is feasible. The L-HRV-S_T_ is the most promising HRV-based objective metric of team collaboration. Analysis of HRV synchrony shows that LOP is shorter when a surgical pair is more synchronised during the procedure. Intraoperative STAI scores of each team member as well as NOTECHS score seem not to be correlated with HRV synchrony metrics. Experience of each team member has a differing relationship with HRV synchrony and LOP. The real-world exploratory nature of the data presented and subsequent analysis is both a strength and weakness in terms of the strength of conclusions we can draw. We suggest that further work may clarify and delineate the interesting associations we report, with the aim of eventually being able to objectively measure teamwork.

## Figures and Tables

**Figure 1 sensors-22-08998-f001:**
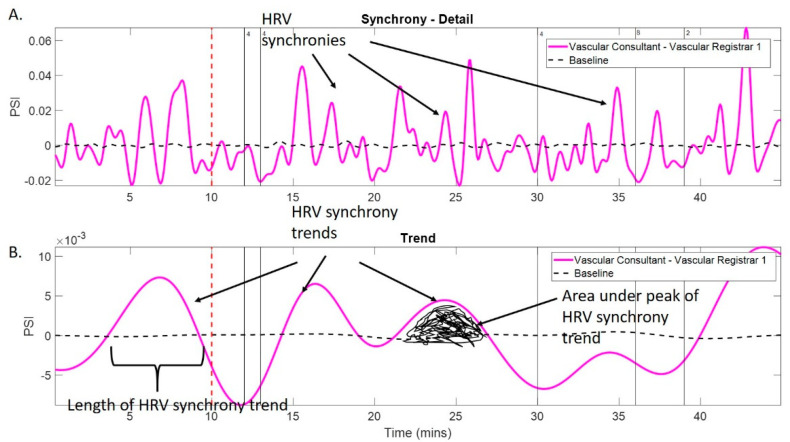
Example of synchrony analysis in a vascular surgical pair. (**A**) shows detailed HRV synchrony (magenta) for the duration of the procedure, black arrows show HRV synchronies; (**B**) shows the trend of HRV synchrony for that procedure (magenta), the baseline synchrony is also shown (black, dotted), HRV synchrony trends are marked with black arrows, length of the HRV synchrony trend is marked with a black bracket and area under the peak of the HRV synchrony trend is marked in black.

**Figure 2 sensors-22-08998-f002:**
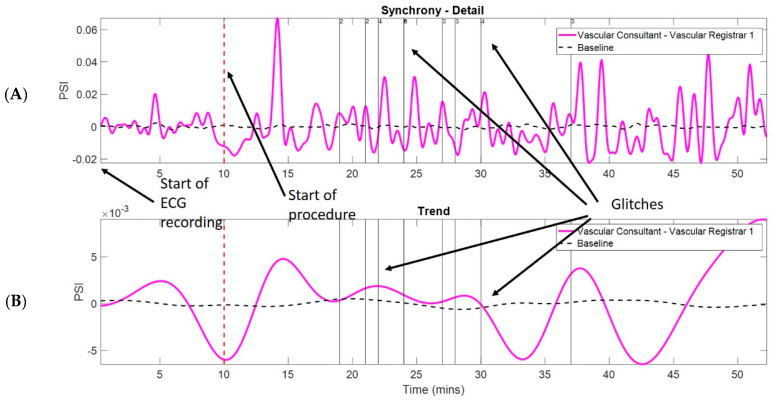
Heart rate variability (HRV) synchronies and synchrony trends within vascular pair in response to glitches from a representative case. Detailed HRV synchrony (**A**) and its trend (**B**) (magenta) are shown. Black dotted horizontal line shows baseline of HRV synchrony. Glitches are shown as vertical black solid lines with the number of a recorded glitch at the top (black arrows showing glitches). Red vertical dotted line shows the time of starting the procedure.

**Figure 3 sensors-22-08998-f003:**
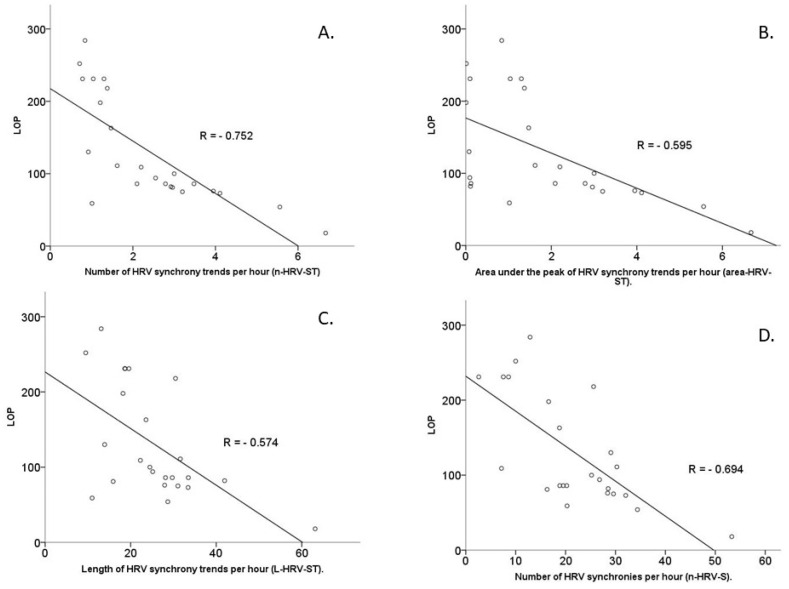
Correlations between LOP and number of HRV synchrony trends per hour (**A**), area under peak of HRV synchrony trends per hour (**B**), length of HRV synchrony per hour (**C**) and number of HRV synchronies per hour (**D**).

**Table 1 sensors-22-08998-t001:** Demographic data for surgical teams.

	Combined (ENT and Vascular)	ENT	Vascular
Experience of consultant (years) (median, (IQR))	35 (32–35)	35 (35–35)	32 (24.5–32)
Experience of first registrar (years) (median, (IQR))	9 (8–10)	8 (8–8)	10 (10–13)
Experience of second registrar (years) (median, (IQR))	4 (3–12)	12 (3–12)	3.5 (3–4)
Team experience (years)—combined sum of years for all team members (median, (IQR))	46 (45–55)	55 (47–55)	45 (38.25–46)
	Composition of Surgical Pairs		
	Frequency (%)	Frequency (%)	Frequency (%)
Consultant–first registrar pair	10 (41.7%)	3 (21.4%)	7 (58.3%)
Consultant–second registrar pair	7 (29.2%)	6 (42.9%)	1 (8.3%)
First registrar–second registrar pair	7 (29.2%)	3 (21.4%)	4 (33.3%)

**Table 2 sensors-22-08998-t002:** Descriptive analysis of HRV synchrony metrics and teams’ data.

	Total	ENT	Vascular
STAI score for consultant—preoperative (median, (IQR))	9 (7–11.75)	10.5 (7–12)	8.5 (7.25–9.75)
STAI score for first registrar—preoperative (median, (IQR))	12.5 (10.25–14.75)	13.5 (9.5–15)	12 (10.25–14)
STAI score for second registrar—preoperative (median, (IQR))	9.5 (7–13)	10 (8–12.75)	9.5 (7–15)
Team STAI score—preoperative—combined scores from all team members (median, (IQR))	35 (26.25–36.75)	35 (28.25–35.75)	31.5 (25.25–38)
STAI score for consultant—intraoperative (median, (IQR))	12.5 (8.25–13)	12.5 (9–13)	12.5 (6.5–13.75)
STAI score for first registrar—intraoperative (median, (IQR))	13.5 (9.25–16.5)	13.5 (9.25–16.5)	13.5 (9.75–16.25)
STAI score for second registrar—intraoperative (median, (IQR))	10 (7–13.75)	10 (6–11)	10 (7–16)
Team STAI score—intraoperative—combined scores for all team members (median, (IQR))	33.5 (28.5–41)	33.5 (28.25–40.5)	33.5 (28.5–43)
NOTECHS score (median, (IQR))	35.5 (24.75–40.75)	40.5 (32.5–43.5)	27.5 (21–35.75)
Glitch rate (per hour) (median, (IQR))	4.8 (3.6–8.3)	4.9 (3.2–8.3)	4.8 (3.7–8.4)
Familiarity Score (median, (IQR))	8.3 (7.4)	9.0 (7.4)	7.60 (10.50)
Number of HRV synchronies per hour (n-HRV-S) (median, (IQR))	20.26 (13.73–28.93)	27.61 (20.45–30.10)	14.59 (7.79–23.89)
Length of HRV synchrony trends per hour (L-HRV-S_T_) (median, (IQR))	24.81 (12.59)	30.38 (7.62)	18.65 (8.85)
Area under the peak of HRV synchrony trend per hour (area-HRV-S_T_) (median, (IQR))	1.42 (2.88)	2.44 (3.42)	1.17 (1.93)
Number of HRV synchrony trends per hour (n-HRV-S_T_) (median, (IQR))	2.15 (2.07)	2.96 (1.62)	1.25 (1.16)

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
