# Peer review of "A Pilot Study of Heart Rate Variability Synchrony as a Marker of Intraoperative Surgical Teamwork and Its Correlation to the Length of Procedure"

_sensors, 2022, doi:10.3390/s22228998_

Round 1

Reviewer 1 Report

The study by Katarzyna Powezka et al. is an interesting study that investigates whether heart rate variability synchrony can be used as an objective marker of teamwork in surgical teams. The manuscript is well-written and can be of interest to the readership of sensors. The authors emphasize that they report the results of a pilot study and carefully discuss the limitations of the study design. However, I believe that some additional aspects should be addressed before the manuscript is considered for publication.

Major concerns:

 The authors analysed the data of 24 surgical pairs. However, it is not mentioned, how many surgeons were indeed involved in the study. This information is relevant for the proper interpretation of the results, so, please include this data, too (e.g in Table 1.)

In my opinion, the interpretation of the correlation between surgeons’ experience and length of procedure is controversial due to the (probably) low number of surgeons involved. Figure 3 shows that that experience of the 1st registrar is positively, whereas the experience of the second registrar is negatively associated with LOP. In my opinion, these conclusions are not valid. Looking at Figure 3A, it appears that there was a first registrar with 10, and one with 14 years of experience. Both participated in longer procedures. This may be due to the fact that they were involved in more complex, longer procedures because they had more experience. The same bias applies to Figure 3B. Presumably there was a single surgeon with 12 years of experience. Without his/her results the relationship would be completely different. The authors also concluded that the experience of the first registrar was inversely correlated with certain HRV synchrony parameters. This may also be an invalid conclusion for the aforementioned reasons. Therefore, I recommend not to include the experience of individual team members in the multiple linear regression models, because you may end up with invalid conclusions. On the other hand, I agree that experience of team members is an important variable in the models, but as it is a pilot study with small number of participants, I suggest that only  the parameter “combined team’s experience” should be included. Therefore, I propose to revise the regression models, figures, and conclusions by excluding the experience of individual team members.  

Minor concerns:

Please also provide the sampling frequency of the ECG you used. This is relevant for HRV analysis.

Please review the abbreviations in the text, not all are explained at first mention.

Reviewer 2 Report

1. The texts in the figures are too small to read. Can you have another look to see if figures and tables could not be better placed, and better formatted?

2. The authors should improve related work and provide an in-depth literature review (with a comprehensive explanation and highlighting the limitations of those).

3. Incomplete information in some references.

4. Results part needs to be improved, the description and performance comparison needs more details.

Round 2

Reviewer 1 Report

The authors have addressed the major concerns I raised. They have removed results which could not be validly interpreted in this pilot study. I recommend publication of the manuscript.

My only advise is to increase the font size of numbers and letters in the graphs of „Figure 3” for better clarity.